# Simulation Studies on Single-Event Effects and the Mechanisms of SiC VDMOS from a Structural Perspective

**DOI:** 10.3390/mi14051074

**Published:** 2023-05-18

**Authors:** Tao Liu, Yuan Wang, Rongyao Ma, Hao Wu, Jingyu Tao, Yiren Yu, Zijun Cheng, Shengdong Hu

**Affiliations:** 1School of Microelectronics and Communication Engineering, Chongqing University, Chongqing 400044, China; 2Science and Technology on Analog Integrated Circuit Laboratory, Chongqing 401332, China

**Keywords:** silicon carbide (SiC), vertical diffuse metal-oxide-semiconductor field transistor (VDMOS), trench, superjunction (SJ), single-event effect (SEE), single-event transient (SET), single-event burnout (SEB), single-event gate rupture (SEGR), linear energy transfer (LET), charge enhancement factor (CEF)

## Abstract

The single-event effect reliability issue is one of the most critical concerns in the context of space applications for SiC VDMOS. In this paper, the SEE characteristics and mechanisms of the proposed deep trench gate superjunction (DTSJ), conventional trench gate superjunction (CTSJ), conventional trench gate (CT), and conventional planar gate (CT) SiC VDMOS are comprehensively analyzed and simulated. Extensive simulations demonstrate the maximum SET current peaks of DTSJ−, CTSJ−, CT−, and CP SiC VDMOS, which are 188 mA, 218 mA, 242 mA, and 255 mA, with a bias voltage *V*_DS_ of 300 V and LET = 120 MeV·cm^2^/mg, respectively. The total charges of DTSJ−, CTSJ−, CT−, and CP SiC VDMOS collected at the drain are 320 pC, 1100 pC, 885 pC, and 567 pC, respectively. A definition and calculation of the charge enhancement factor (CEF) are proposed. The CEF values of DTSJ−, CTSJ−, CT−, and CP SiC VDMOS are 43, 160, 117, and 55, respectively. Compared with CTSJ−, CT−, and CP SiC VDMOS, the total charge and CEF of the DTSJ SiC VDMOS are reduced by 70.9%, 62.4%, 43.6% and 73.1%, 63.2%, and 21.8%, respectively. The maximum SET lattice temperature of the DTSJ SiC VDMOS is less than 2823 K under the wide operating conditions of a drain bias voltage *V*_DS_ ranging from 100 V to 1100 V and a LET value ranging from 1 MeV·cm^2^/mg to 120 MeV·cm^2^/mg, while the maximum SET lattice temperatures of the other three SiC VDMOS significantly exceed 3100 K. The SEGR LET thresholds of DTSJ−, CTSJ−, CT−, and CP SiC VDMOS are approximately 100 MeV·cm^2^/mg, 15 MeV·cm^2^/mg, 15 MeV·cm^2^/mg, and 60 MeV·cm^2^/mg, respectively, while the value of *V*_DS_ = 1100 V.

## 1. Introduction

Recently, silicon carbide (SiC) power devices have gained more and more popularity in the automotive, renewable energy, high-speed railway, and aerospace industries, due to their outstanding comprehensive characteristics, such as a higher breakdown voltage (BV), higher thermal conductivity, lower specific ON-resistance (*R*_on,sp_), and so on [1,2], compared with silicon power devices. The SiC vertical diffuse metal-oxide-semiconductor field transistor (VDMOS) is one of the most widely used power devices, due to its merits of high input impedance, fast switching, easy driving, etc.

The power device is one of the most indispensable components in aerospace systems. SiC material is regarded as one of the best candidates for aerospace applications due to its higher electron-hole pair generation energy of 7.8–9 eV and higher displacement energy of 20–35 eV, compared with its silicon counterparts (3.6 eV and 13–20 eV, respectively) [3]. However, heavy ion, neutron, and proton irradiations have demonstrated that SiC power devices (Schottky diodes, MOSFETs, and insulated-gate bipolar transistors) are highly susceptible to heavy ion or particle irradiation, which may cause catastrophic damage such as single-event burnout (SEB) [4,5,6,7,8,9,10] and single-event gate rupture (SEGR) [11]. Experiments have demonstrated that neutrons could induce SEB failure in SiC VDMOS and that failures are dependent on reverse-gate bias [12,13,14]. Heavy ion irradiation could also lead to leakage current in SiC VDMOS via heavy ion irradiation [15,16,17]. Protons could also trigger SEB in the SiC VDMOS [8].

The SEB mechanisms have gradually gained more and more interest over the past few years in the context of parasitic bipolar junction transistor turn-on mechanisms, their contributions to SEB having been reported in several studies [5,18]. Nonetheless, in contrast to silicon power devices, research studies on SEE and the mechanism of SiC VDMOS are relatively few; in general, such extensive research is essential and worthwhile.

The reported studies primarily focused on 1200 V SiC VDMOS [5,9,13,15]. The single-event effect (SEE) mechanisms of novel structure deep-trench SiC VDMOS proposed by the authors with a BV exceeding 2500 V, another two kinds of conventional trench structures with a BV exceeding 2000 V, and a conventional planar structure with a BV of 1200 V are comparatively investigated in this paper.

The device structures and basic performances are simulated and summarized in Section 2. The single-event transient (SET) current, lattice temperature mechanisms and electric field are simulated and analyzed in Section 3, where the research results are also presented. Our conclusions are offered in Section 4.

## 2. Device Structures and Performances

The cross-sectional half-cell of the DTSJ SiC VDMOS proposed by the authors of [19], CTSJ SiC VDMOS, CT SiC VDMOS, and CP SiC VDMOS, as shown in Figure 1, are selected for SEE investigation. The primary geometry and doping parameters for each SiC VDMOS are listed in Table 1. The epitaxial thickness of DTSJ SiC VDMOS is designed to be 18 μm, targeting a BV of 2500 V, then the epitaxial thickness of the other three structures of SiC VDMOS is also chosen to be 18 μm for comparison. The N-pillar and P-pillar have the same width and doping concentration in the superjunction (SJ) structure to ensure charge balance and achieve the maximum BV [20,21]. The doping concentration in the epitaxial layer for each SiC VDMOS is optimized by maximizing the figure of merit (FOM, defined as *BV*^2^/*R*_on,sp_, where *R*_on,sp_ is defined as *W* × 1 μm × *V*_DS_/*I*_DS_, and *V*_DS_ = 5 V is chosen, based on the consideration of the large signal condition in switching mode for VDMOS), as shown in Figure 2; the simulation results show the downward parabola relationship between FOM and doping concentration. In addition, it clearly demonstrates that the DTSJ SiC VDMOS has the highest FOM and BV and the lowest *R*_on,sp_. The optimized doping concentration in the drift or SJ of DTSJ−, CTSJ−, CT−, and CP SiC VDMOS is 5.6 × 10^16^ cm^−3^, 2.8 × 10^16^ cm^−3^, 5.2 × 10^15^ cm^−3^, and 7 × 10^15^ cm^−3^, respectively. The doping concentration of CP SiC VDMOS is higher than that of CT SiC VDMOS, to obtain a reasonable *R*_on,sp_.

The breakdown characteristics of the four SiC VDMOS devices are given in Figure 3a. The BV of DTSJ−, CTSJ−, CT−, and CP SiC VDMOS are 2510 V, 2405 V, 2255 V, and 1248 V, respectively. The output characteristic *I*_ds_-*V*_ds_, with a fixed gate bias voltage, *V*_GS_, of 20 V and a drain bias voltage *V*_DS_ ranging from 0 to 5 V, is given in Figure 3b; the ON-resistance at a *V*_DS_ = 5 V of DTSJ−, CTSJ−, CT−, and CP SiC VDMOS are 1.19 mΩ·cm^2^, 1.42 mΩ·cm^2^, 2.38 mΩ·cm^2^, and 11.00 mΩ·cm^2^, respectively. The FOM of DTSJ−, CTSJ−, CT−, and CP SiC VDMOS are 5294 MW·cm^2^, 4073 MW·cm^2^, 2136 MW·cm^2^, and 141 MW·cm^2^, respectively.

## 3. Results and Discussion

### 3.1. SEE Simulation Setups and Models

In this work, the simulations show that the lattice temperature significantly influences the SET current. As shown in Figure 4a, the simulated SET current of the CP SiC VDMOS increases immediately after heavy ions strike the device, with a drain bias voltage *V*_DS_ of 500 V, a linear energy transfer (LET) of 120 MeV·cm^2^/mg, and an ion penetration depth of 20 μm, without the lattice temperature being taken into consideration. It then rapidly reaches a steady current of approximately 1.15 A. In contrast, the SET current behaves according to the double exponential law when the lattice temperature is involved in the simulation. First, the SET current increases rapidly and achieves a peak value of 300 mA within 200 pS, which is somewhat consistent with the heavy ion measurement results in terms of magnitude [12], then it decays exponentially. As shown in Figure 4b, the impact ionization rate with the lattice temperature turned on is larger than when the lattice temperature is turned off. Meanwhile, as shown in Figure 4c, electron mobility when the lattice temperature is turned on is also slower than that without the lattice temperature being involved. The electron mobility keeps almost constant at 150 cm^2^/V·S in the whole drift region without the involvement of lattice temperature, while the electron mobility varies exponentially with vertical distance and decreases to less than 30 cm^2^/V·S near the interface of drift and substrate, which causes the velocity of the carriers to decrease dramatically with an increase in lattice temperature; as a consequence, the total terminal current decreases. This demonstrates that the lattice temperature must be considered in the SEE simulation of SiC VDMOS with TCAD tools.

In a word, for accurate simulation purposes, models of carrier generation and recombination (SRH, Auger), mobility (DopingDependence, HighFieldSaturation, Enormal), lattice temperature, avalanche multiplication (OkutoCrowell), and thermal effects should all be taken into consideration in the SEE simulation of a SiC VDMOS, using the Sentaurus TCAD tool.

The heavy ion model embedded in the Sentaurus TCAD tool is shown in Figure 5. As shown in Figure 5a, *l*_max_ is the length of the ion track. The e-h pair generation rate is defined using Formulas (1) and (2), where *R*(w) and *T*(t) are functions describing the spatial and temporal variations of the generation rate, and *G_LET_*(*l*) is the linear energy transfer generation density [22].
(1)$$G(l,w,t)=G_{LET}(l)R(w,l)T(t)\left(l<l_{max}\right)$$
(2)$$G(l,w,t)=0\left(l>l_{max}\right)$$

An example is shown in Figure 5b; the spatial distribution is Gaussian, the unit of *LET_f* is pC/μm, the unit of *LET* is MeV·cm^2^/mg, and the conversion relationship between *LET_f* and *LET* is given in Formula (3).
(3)LET_f=0.007×LET

The SEE simulation involves the heavy ion striking locations, direction, energy, the species of heavy ion, as well as the bias voltage on the device. As shown in Figure 6a,b, three regions, comprising body contact (A), source (B), and oxide (C), are chosen as the ion-striking locations for the trench gate devices. As shown in Figure 6b, four regions, comprising body contact (A), source (B), channel (D), and JFET region (C), are chosen as the heavy ion striking locations for planar one. The incidence of heavy ions is considered to be normal, for extensive simulation convenience. The simulation matrix for the bias voltage and LET is shown in Figure 6d; the bias voltage on SiC VDMOS is the same as in the BV simulation shown in Figure 6c, namely, *V*_G_ = *V*_S_ = 0, where the drain bias voltage *V*_DS_ ranges from 100 V to 1100 V with a step of 200 V, the LET ranges from 1 MeV·cm^2^/mg to 120 MeV·cm^2^/mg. It needs to be pointed out that small LET values of 1 MeV·cm^2^/mg, 3 MeV·cm^2^/mg, 6 MeV·cm^2^/mg, 9 MeV·cm^2^/mg, and 12 MeV·cm^2^/mg are chosen for the SEB threshold simulation.

### 3.2. Model Validation by Comparison with Heavy Ion Experiment Results

To validate the TCAD model, the TCAD simulated SEB-threshold *V*_DS_ voltages for 1200 V CP SiC VDMOS were compared with the heavy ion results measured by Witulki [5], Lauenstein [9,10], and Pengwei Li [15], as shown in Figure 7. The BV of all the samples for the heavy ion test is about 1200 V and all the samples are of the planar gate SiC VDMOS. The ion species exploited in the heavy ion test include B, N, Ne, Ar, Cu, Kr, Ag, and Xe, and the corresponding LETs of the ions are approximately 1 MeV·cm^2^/mg, 2 MeV·cm^2^/mg, 3.9 MeV·cm^2^/mg, 10 MeV·cm^2^/mg, 23 MeV·cm^2^/mg, 40 MeV·cm^2^/mg, 49 MeV·cm^2^/mg, and 73 MeV·cm^2^/mg. Thus, the SEB-threshold *V*_DS_ voltages with LETs exploited in the experiments for 1200 V CP SiC VDMOS were simulated; the simulated results are shown as a solid blue line in Figure 7. The SEB threshold *V*_DS_ voltage declines rapidly with an LET value of less than 10 MeV·cm^2^/mg and decreases gradually with an LET value larger than 10 MeV·cm^2^/mg, which is similar to the simulation performed by Witulki [5].

The SEB threshold *V*_DS_ voltages for the B and Ar ion striking were about 1100 V and 600 V, as taken from Witulki [5], while the simulated ones were 1190 V and 680 V, and the relative error tolerances were less than 15%. The SEB threshold *V*_DS_ voltages for Cu and Xe ion striking were about 550 V and 600 V, as taken from Lauenstein [9,10], while the simulated ones were 675 V and 658 V and the relative error tolerances were less than 20%.

There are certain deviations between the simulation results and measured results because of the differences in epitaxial thickness, doping concentration, etc. The geometric and doping parameters of these commercial devices are not available publicly.

The TCAD simulation results agree with some of the heavy ion experimental results to some extent, indicating that the TCAD model works effectively.

### 3.3. SET Current and Mechanisms

The physical process involving energetic heavy ions or particles interacting with a device can be conceptually understood with the help of Figure 8. It is generally recognized that four steps happen when a heavy ion or a particle strikes the VDMOS, as illustrated in Figure 8. Firstly, the heavy ion or particle penetrates the device and collides with the lattice, atoms, and other carriers; the energy decreases gradually and is transferred into the device concurrently. The energy loss is defined using linear energy transfer (LET: MeV/mg/cm^2^ or MeV·cm^2^/mg), as shown in Figure 8a. Secondly, a considerable number of electron and hole pairs (e-h pairs) are produced along the trajectory of the heavy ion or particle; the quantity of the e-h pairs is closely correlated with the species, range, energy, and incidence angle of the heavy ion or particle, as shown in Figure 8b. Thirdly, the induced e-h pairs drift under the electric field *E* from the reverse bias voltage of the drain, and electrons move toward the drain region and holes toward the source region, as shown in Figure 8c. Finally, the induced electrons are collected at the drain, and the holes collected at the source, causing an additional electric field, *E*_ADD,_ equivalently, further disturb the potential of the drain. Consequently, this may trigger the conduction of BJT or SCR structures in power devices and result in catastrophic single–event burnout [6,23]. 

The simulated SET current pulse results, with an ion striking location X of 1.5 μm, a LET of 60 MeV·cm^2^/mg, *V*_DS_ of 300 V, and a penetration depth ranging from 0.5 μm to 20 μm, are shown in Figure 9. The SET current increases immediately after the ion strikes in several picoseconds, then it decreases. The width of the pulse is typically from tens of picoseconds to several nanoseconds. Both the pulse width and the SET current pulse peak increase with the heavy ion penetration depth, the peak value attaining as high as 150 mA with an ion depth of 20 μm. Interestingly, there is a plateau with a depth ranging from 3 to 10 μm, while the SET current decreases steeply for a depth larger than 15 μm. This phenomenon can be explained by gaining an insight into the characteristics of electrons and holes, as shown in Figure 10 and Figure 11. Both the induced e-h densities are 6.5 × 10^18^/cm^2^ along the ion trajectory for depths of 0.5 μm and 1.2 μm, which corresponds to the heavy ion striking directly in the body region. In contrast, the induced e-h pair density ranges from 4 × 10^18^/cm^2^ to 8.8 × 10^18^/cm^2^ in the whole drift region for a depth of 20 μm. The velocities of the electrons are approximately 1 × 10^6^ cm/S, 1 × 10^6^ cm/S, 1.6 × 10^17^ cm/S for depths of 0.5 μm, 1.2 μm, and 20 μm, respectively. The velocities of the holes are approximately 1 × 10^5^ cm/S, 1 × 10^5^ cm/S, and 3 × 10^16^ cm/S for depths of 0.5 μm, 1.2 μm, and 20 μm, respectively. According to the current density law of semiconductors, the SET current should be largest for a depth of 20 μm.

Logically, the charge generated by heavy ion striking should be linearly dependent on ion properties, such as LET, energy, incidence direction, penetration depth, and the internal electric field from reverse bias voltage. However, due to the synergistic effect of the semiconductor’s periodic potential field, carrier generation and recombination, and external bias voltage, in fact, the charge collected at the high drain bias or LET is much greater than expected and can be defined as SEE charge enhancement. 

A large electric field, *E*, exits in the drift region with a drain reverse bias voltage; the induced e-h pairs drift along the electric field, the speed of which is growing faster and faster, the accelerated e-h pairs collide with lattice or other carriers, and more and more electrons and holes are further generated in this procedure. In this cycled process, numerous electrons and holes are generated via avalanche multiplication. As shown in Figure 12, the total charge generated by the heavy ions linearly increases with depth; the total charge increase is not always linear but is exponential since the LET value is larger than 15 MeV·cm^2^/mg. 

In this work, in order to characterize the e-h pairs avalanche multiplication quantitatively, the charge enhancement factor (*CEF*) of e-h pairs is defined as follows:(4)$$CEF=\frac{Q_{simulated}-Q_{Linear,extra}}{Q_{Linear,extra}}$$
where *Q_simulated_* denotes the total charge collected at the drain terminal by an integral of the current, and *Q_Linear,extra_* represents the total charge obtained by linear extrapolation, as illustrated in Figure 12.

The total charge-generated dependence on heavy ion penetration depth for DTSJ−, CTSJ−, CT−, and CP SiC VDMOS is shown in Figure 12a–d, with an ion striking location, X, of 1.5 μm, an LET value of 60 MeV·cm^2^/mg, and a drain bias voltage *V*_DS_ of 300 V. For a depth of 20 μm, the total charges collected at the drain for DTSJ−, CTSJ−, CT−, and CP SiC VDMOS are 320 pC, 1100 pC, 885 pC, and 567 pC, respectively. Meanwhile, the charge enhancement factors for DTSJ−, CTSJ−, CT−, and CP SiC VDMOS are 43, 160, 117, and 55, respectively.

### 3.4. Sensitive Volume Analysis

Assuming a heavy ion penetration depth of 1.5 μm, a drain bias voltage *V*_DS_ of 300 V, and an LET value ranging from 1 MeV·cm^2^/mg to 120 MeV·cm^2^/mg, the simulated SET currents of four SiC VDMOS are shown in Figure 13. The SET current feature for four SiC VDMOS is a plateau in the current pulse.

As shown in Figure 13a, the pulse widths of the DTSJ SiC VDMOS are approximately 1.1 nS, and 0.6 nS for a striking location at the source (X = 1.5 μm) and P-body (X = 0.5 μm); the mean currents in the plateau are about 0.8 mA and 0.78 mA for striking locations at the source (X = 1.5 μm) and P-body (X = 0.5 μm). In contrast, the pulse width is less than 0.3 nS for the striking location at the gate (X = 2.5 μm).

As shown in Figure 13b, the pulse widths of the CTSJ SiC VDMOS are approximately 1.1 nS, and 0.7 nS for a striking location at the source (X = 1.8 μm) and P-body (X = 0.5 μm), while the mean currents in the plateau are about 0.8 mA and 0.6 mA for striking locations at the source (X = 1.5 μm) and P-body (X = 0.5 μm). In contrast, the pulse width is less than 0.3 nS for the striking location at the gate (X = 2.5 μm).

As shown in Figure 13c, the pulse widths of the CT SiC VDMOS are approximately 1.7 nS, and 1.15 nS for a striking location at the source (X = 1.5 μm) and P-body (X = 0.5 μm), while the mean currents in the plateau are about 0.7 mA and 0.7 mA for striking locations at the source (X = 1.5 μm) and P-body (X = 0.5 μm). The pulse width is less than 0.1 nS for the striking location at the gate (X = 2.5 μm).

As shown in Figure 13d, the pulse widths of CP SiC VDMOS are approximately 1.3 nS, 2.2 nS, 3.4 nS, and 3.3 nS for the striking locations at the P-body (X = 0.4 μm), source (X = 1.2 μm), channel (X = 2 μm), and JFET region (X = 2.7 μm), respectively, while the mean currents in the plateau are all about 0.93 mA for the striking locations at the P-body (X = 0.4 μm), source (X = 1.2 μm), channel (X = 2 μm), and JFET region (x = 2.7 μm), respectively. 

According to the simulation, the volumes under the source are more sensitive than those for the trench gate SiC VDMOS, while those under the trench gate are the least sensitive. The volume in the channel region is the most sensitive for the planar gate SiC VDMOS, while the volumes under the source region are the most sensitive for the trench gate SiC VDMOS. 

Anyway, compared with the trench structure devices, the whole device of the planar structure SiC VDMOS is relatively more susceptible to heavy ion radiation.

At a heavy ion penetration depth of 20 μm, drain bias voltage *V*_DS_ of 300 V, and LET ranging from 1 MeV·cm^2^/mg to 120 MeV·cm^2^/mg, the simulated SET current of four SiC VDMOS samples are shown in Figure 14. Firstly, as far as the DTSJ SiC VDMOS is concerned, the most sensitive regions are the under-body contact (X = 0.5 μm) and source (X = 1.5 μm), the SET current exceeds 150 mA and is less than 200 mA, with an LET value larger than 15 MeV·cm^2^/mg, the region under oxide (X = 2.5 μm) is less sensitive, and the SET current is almost zero. Secondly, for CTSJ SiC VDMOS, it seems that all regions are more sensitive, with a higher LET that is larger than 15 MeV·cm^2^/mg, while the SET current exceeds 180 mA and is less than 220 mA. In addition, all the regions in both CT− and CP SiC VDMOS are similar to CTSJ SIC VDMOS when the energetic heavy ion LET is larger than 15 MeV·cm^2^/mg. Thirdly, the feature for four kinds of devices is that the LET threshold, defined as the point where the SET current sharply rises, is approximately 15 MeV·cm^2^/mg.

Based on the above simulation results, it could be inferred that the SET current for the trench gate device is less than that of the conventional planar device, while the SET current for the SJ device is slower than that of the non-SJ device; thus, the proposed novel DTSJ SiC VDMOS has the best SET current characteristics.

### 3.5. SEB and Thermal Analysis

The simulated SET lattice temperature evolution over time of DTSJ SiC VDMOS, with a heavy ion striking location X = 1.5 μm, LET of 60 MeV·cm^2^/mg, drain bias voltage *V*_DS_ of 300 V, and a penetration depth ranging from 0.5 μm to 20 μm, are shown in Figure 15. As with the SET current, the lattice temperature also behaves in the same way as in the double exponential law. The lattice temperature peak is less than 350 K when the depth is less than 5 μm, it is less than 550 K when the depth is less than 15 μm, and the peak is as high as 1980 K when the depth is larger than 20 μm. The mechanisms can be explained further by gaining an insight into the power distribution along the heavy ion trajectory when the lattice temperature reaches the peak, as shown in Figure 16. The peak hole current densities are about 1.2 × 10^5^ A·cm^2^, 1.3 × 10^5^ A·cm^2^, and 25 × 10^5^ A·cm^2^ for depths of 1.2 μm, 5 μm, and 20 μm, respectively. The hole drifts along the electric field; therefore, the hole current densities fall with depth. The peak electron current densities are about 6.5 × 10^5^ A·cm^2^, 8.5 × 10^5^ A·cm^2^, and 16 × 10^6^ A·cm^2^ for depths of 1.2 μm, 5 μm, and 20 μm, respectively. The power density distribution is shown in Figure 16c; the power density peaks are about 6 × 10^7^ A·cm^2^, 6.4 × 10^7^ A·cm^2^, and 30 × 10^8^ A·cm^2^ for depths of 1.2 μm, 5 μm, and 20 μm, respectively. The maximum power density of DTSJ SiC VDMOS with an ion depth of 20 μm is up to 50 times that with a depth of 5 μm, while the location is close to the interface of the N-pillar and N+ substrate (y = 18 μm), which indicates probable SEB occurrence at this point.

Extensive simulations have been carried out to explore the SET temperatures in the four SiC VDMOS devices, with comprehensive LET values ranging from 1 MeV·cm^2^/mg to 120 MeV·cm^2^/mg and drain bias voltage *V*_DS_ values ranging from 100 V to 1100 V with a step of 200 V, where a LET value as small as 1 MeV·cm^2^/mg is designed to obtain the threshold LET value. The simulated SET lattice temperature peaks of DTSJ−, CTSJ−, CT−, and CP SiC VDMOS are shown in Figure 17.

For the DTSJ SiC VDMOS, as shown in Figure 17a, in the case of a drain bias voltage *V*_DS_ = 100 V, the lattice temperature is lower than 400 K when the LET value is less than 45 MeV·cm^2^/mg, and the lattice temperature increases with LET. It begins to saturate when the LET value is greater than 75 MeV·cm^2^/mg; the higher the drain bias voltage *V*_DS_, the smaller the threshold of LET where the lattice temperature begins to saturate, and the higher the saturation value of the lattice temperature. The saturation values of the lattice temperature are 1670 K, 2190 K, 2229 K, 2443 K, 2628 K, and 2823 K, and the LET threshold values are approximately 75 MeV·cm^2^/mg, 9 MeV·cm^2^/mg, 3 MeV·cm^2^/mg, 3 MeV·cm^2^/mg, 1 MeV·cm^2^/mg, and 1 MeV·cm^2^/mg, corresponding to the drain bias voltages of 100 V, 300 V, 500 V, 700 V, 900 V, and 1100 V.

For CTSJ SiC VDMOS, as shown in Figure 17b, the saturation values of lattice temperature are 1553 K, 2249 K, 2487 K, 3132 K, >3200 K, and >3500 K, respectively. Incidentally, the melting point of the SiC material is about 3100 K [24], while the LET thresholds are approximately 75 MeV·cm^2^/mg, 12 MeV·cm^2^/mg, 6 MeV·cm^2^/mg, 3 MeV·cm^2^/mg, 1 MeV·cm^2^/mg, and 1 MeV·cm^2^/mg, corresponding to the drain bias voltages of 100 V, 300 V, 500 V, 700 V, 900 V, and 1100 V.

For CT SiC VDMOS, as shown in Figure 17c, the saturation values of lattice temperature are 1550 K, 2227 K, 2700 K, 2968 K, >3500 K, and >3500 K, respectively. The LET thresholds are 75 MeV·cm^2^/mg, 13 MeV·cm^2^/mg, 6 MeV·cm^2^/mg, 3 MeV·cm^2^/mg, 1 MeV·cm^2^/mg, and 1 MeV·cm^2^/mg, corresponding to the drain bias voltages of 100 V, 300 V, 500 V, 700 V, 900 V, and 1100 V.

For CP SiC VDMOS, as shown in Figure 17d, the saturation values of lattice temperature are 1680 K, 2268 K, 2550 K, 2746 K, >3500 K, and >3500 K, respectively. The LET thresholds are 30 MeV·cm^2^/mg, 6 MeV·cm^2^/mg, 3 MeV·cm^2^/mg, 3 MeV·cm^2^/mg, 1 MeV·cm^2^/mg, and 1 MeV·cm^2^/mg, corresponding to drain bias voltages of 100 V, 300 V, 500 V, 700 V, 900 V, and 1100 V.

An interesting phenomenon has been identified by observing the spatial temperature distribution in the device illustrated in Figure 18, corresponding to a heavy ion striking location at X = 1.5 μm, a penetration depth of 20 μm, a high bias voltage *V*_DS_ of 1100 V, and the highest LET of 120 MeV·cm^2^/mg. The relatively high temperature was mainly gathered along the ion trajectory for DTSJ SIC VDMOS and CP SiC VDMOS. However, the relatively high temperatures mainly gathered near the corner of the poly-gate for CTSJ SiC VDMOS and CP SiC VDMOS. The zoomed-in diagrams are shown in Figure 19.

As shown in Figure 19a, for CTSJ SiC VDMOS, where the maximum thermal point is located at the intersection of the N-pillar, N+ substrate, and oxide, which is consistent with the power density in Figure 16c, the highest temperature is about 2823 K and the oxide may be melted first; the melting point of the oxide material is approximately 1900 K, which leads to SEB.

As shown in Figure 19b, for CTSJ SiC VDMOS, the maximum thermal point is located at the intersection of the N-pillar, P-body, and oxide; the highest temperature is beyond 3400 K when both the SiC and the oxide will have melted, resulting in SEB.

As shown in Figure 19c, for CT SiC VDMOS, the maximum thermal point is located at the intersection of the N-pillar, P-body, and oxide; the highest temperature exceeds 3500 K when both the SiC and the oxide will have melted, which leads to SEB.

As shown in Figure 19d, for CP SiC VDMOS, the maximum thermal point is located at the center of the N-drift region; the highest temperature achieves a level as high as 5000 K and the SiC has undoubtedly melted. Thus, SEB occurs.

Numerical simulations elucidated the finding that the SEB occurring point for DTSJ SiC VDMOS is probably located at the intersection of the N-pillar, N+ substrate, and oxide; the SEB occurring point for both CTSJ− and CT SiC VDMOS is probably located at the junction of the N-pillar or N-drift, the P-body, and oxide. In contrast, the SEB occurring point for the CP SiC VDMOS mainly lies in the center of the N-drift region. The distinguishing feature of a three-trench-gate SiC VDMOS is that the SEB point is related to the corner of the gate, where the electric field is the strongest.

### 3.6. SEGR and Electric Field Analysis

The electric field is an essential parameter in the design and optimization of power devices, as in the SEE simulation of SiC VDMOS. The electric field distributions in DTSJ−, CTSJ−, CT−, and CP SiC VDMOS are simulated and illustrated in Figure 20, with an LET value of 60 MeV·cm^2^/mg, a drain bias voltage *V*_DS_ of 300 V, a heavy ion striking depth of 20 μm, and an electric field captured at 5 pS after heavy ion striking.

The zoomed-in diagrams where the maximum electric fields occur are shown in Figure 21. The electric field peaks for three kinds of trench SiC VDMOS devices in the oxide regions are located at the intersection of the oxide and corner of the gate. The electric field peak for the CP SiC VDMOS is located at the side of the oxide closest to the gate.

The electric fields under heavy ion striking and without heavy ion striking, with an LET value of 120 MeV·cm^2^/mg and a drain bias voltage *V*_DS_ of 300 V, are compared and plotted in Figure 22. Heavy ion striking leads to an increase in the electric field in the oxide region, when the electric field peak *E*_p1_ at 5 pS after heavy ion striking is about 4 MV/cm, the electric field peak *E*_p2_ without heavy ion striking is about 2.54 MV/cm, and the difference in magnitude between the two is about 1.6 times. The electric field in the N-pillar and substrate regions for the samples with heavy ion striking and without heavy ion striking are almost the same. Moreover, another electric field spike of *E*_p3_ (≈0.6 MV/cm) exists at the intersection of the N-pillar and substrate region in the case of heavy ion striking. 

The evolution of the electric field in the oxide region over time for DTSJ SiC VDMOS with LET values of 30 MeV·cm^2^/mg, 75 MeV·cm^2^/mg, and 120 MeV·cm^2^/mg are shown in Figure 23a, Figure 23b, and Figure 23c, respectively. After heavy ion striking, the electric field increases immediately and achieves a peak after several picoseconds, then it decreases exponentially. The electric field peaks correlate linearly with the LET value and drain bias voltage *V*_DS_. The bigger the *V*_DS_ and LET, the higher the electric field’s peaks. The electric field peak under LET = 120 MeV·cm^2^/mg and *V*_DS_ = 1100 V is about 13 MV/cm, which exceeds the critical electric field of oxide (≈12 MV/cm) [25]; that is, SEGR occurs.

The electric field peaks in the oxide region for DTSJ−, CTSJ−, CT−, and CP SiC VDMOS, under the conditions of LET values ranging from 1 MeV·cm^2^/mg to 120 MeV·cm^2^/mg and a drain bias voltage of 1100 V, are shown in Figure 24. As shown in Figure 24a, no occurrence of SEGR happens with the DTSJ−, CT−, and CP SiC VDMOS when the drain bias voltage is less than 300 V, while SEGR occurs with CTSJ SiC VDMOS when the LET value is larger than 100 MeV·cm^2^/mg. However, the probability of SEGR occurrence increases with increased LET and drain bias voltage *V*_DS._ As shown in Figure 24b, the SEGR LET thresholds are approximately 100 MeV·cm^2^/mg and 60 MeV·cm^2^/mg for DTSJ− and CP SiC VDMOS, respectively. In contrast, the electric field peaks for CTSJ− and CT SiC VDMOS significantly exceed the critical electric field of oxide, while the SEGR LET thresholds for CTSJ− and CT SiC VDMOS are about 15 MeV·cm^2^/mg.

## 4. Conclusions

The SEE effects and mechanisms of the proposed DTSJ SiC VDMOS are investigated in this paper, accompanied by a comparative study of the other three structures. 

Firstly, extensive simulations demonstrate that the proposed DTSJ SiC VDMOS possesses the best SET current, relatively speaking, and has the lowest charge enhancement factor.

Secondly, by exploring the synergistic effect of the electric field and microcosmic motion of e-h pairs, thermal generation and the way that it conversely impacts the velocity of carriers are uncovered. It is easy to conclude that both the SiC and oxide sides are thermally broken down in the trench SiC VDMOS. At the same time, the N-drift region is melted, primarily in the case of the planar SiC VDMOS.

Thirdly, the SEGR occurrence probability increases with the LET and drain bias voltage *V*_DS_. The bigger the *V*_DS_ and LET values, the higher the electric field’s peaks. Simulations demonstrate that the proposed DTSJ SIC VDMOS has a higher SEGR LET threshold. 

Numerical simulations indicate that both the trench and SJ structures improve the SEE characteristics of SiC VDMOS.

The relatively excellent SET, SEB, and SEGR performances make the proposed DTSJ SiC VDMOS a promising candidate for aerospace and nuclear radiation applications.

## Figures and Tables

**Figure 1 micromachines-14-01074-f001:**
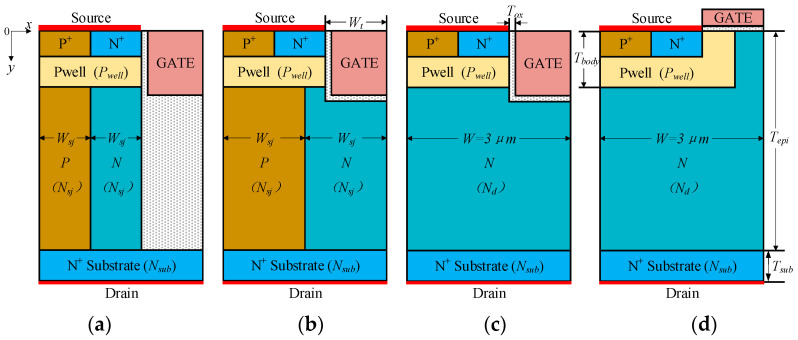
Cross-sectional schematics of the half-cell for (**a**) DTSJ−, (**b**) CT SJ−, (**c**) CT−, and (**d**) CP SiC VDMOS.

**Figure 2 micromachines-14-01074-f002:**
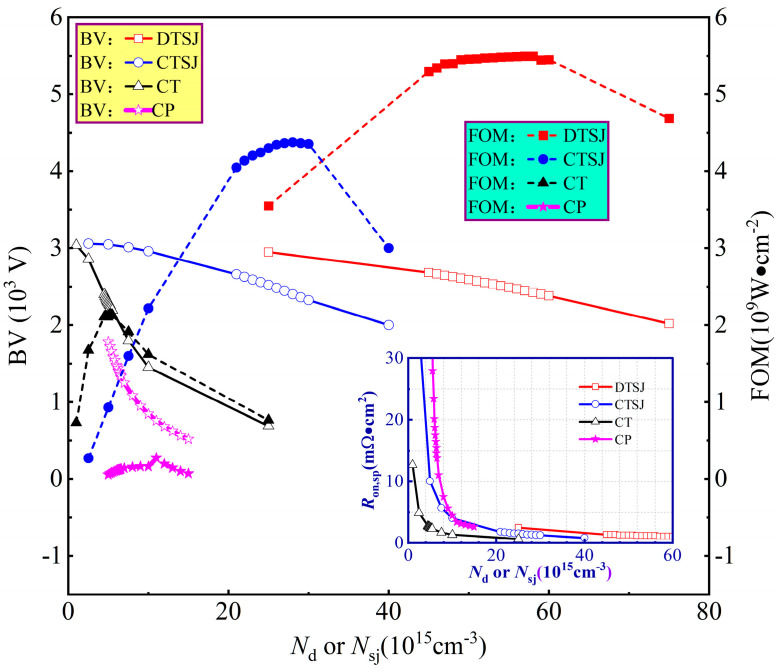
The BV, FOM, and *R*_on,sp_, depending on doping concentration, in the SJ or N- drift region of the DTSJ−, CTSJ−, CT−, and CP SiC VDMOS.

**Figure 3 micromachines-14-01074-f003:**
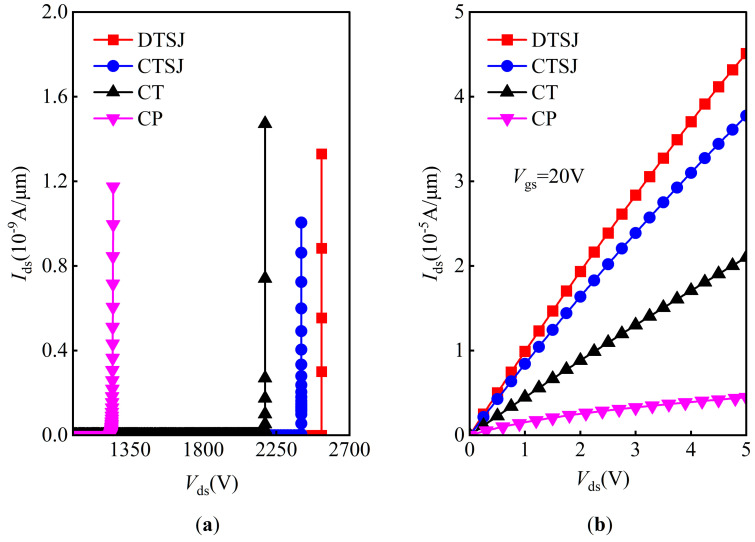
The optimized DC characteristics of the DTSJ−, CTSJ−, CT−, and CP SiC VDMOS. The breakdown criterion is that the drain-source current *I*_ds_ equals 1nA/μm. (**a**) Off-state (*V*_gs_ = 0 V), (**b**) on-state (*V*_gs_ = 20 V).

**Figure 4 micromachines-14-01074-f004:**
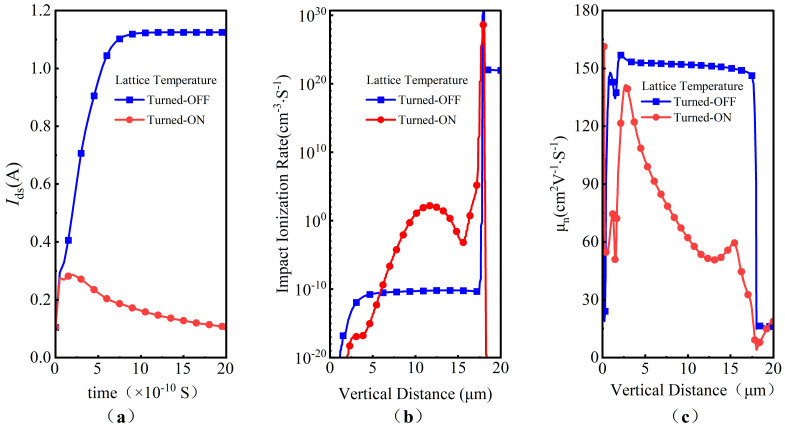
Simulated SEE characteristics of CP SiC VDMOS when the lattice temperature model is turned on and off: (**a**) single-event transient current *I*_ds_; (**b**) impact ionization rate; (**c**) electron mobility.

**Figure 5 micromachines-14-01074-f005:**
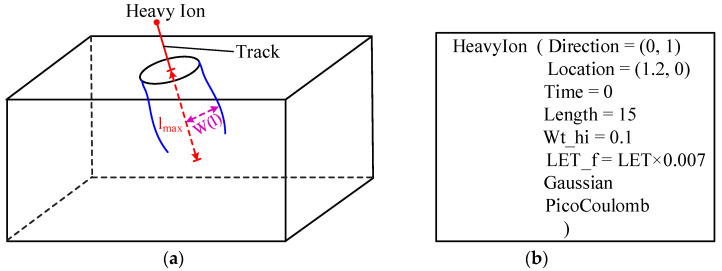
A heavy ion model using the Sentaurus TCAD tool: (**a**) parameters of the ion penetrating device; (**b**) heavy ion model example.

**Figure 6 micromachines-14-01074-f006:**
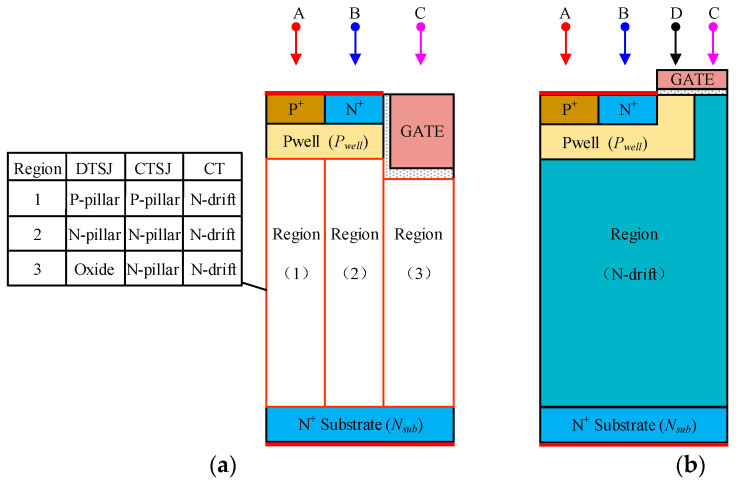
SEE simulation setups: (**a**) ion striking locations for the trench gate SiC VDMOS; (**b**) ion striking locations for the planar gate SiC VDMOS; (**c**) bias voltage for VDMOS; (**d**) the LET and drain bias matrix.

**Figure 7 micromachines-14-01074-f007:**
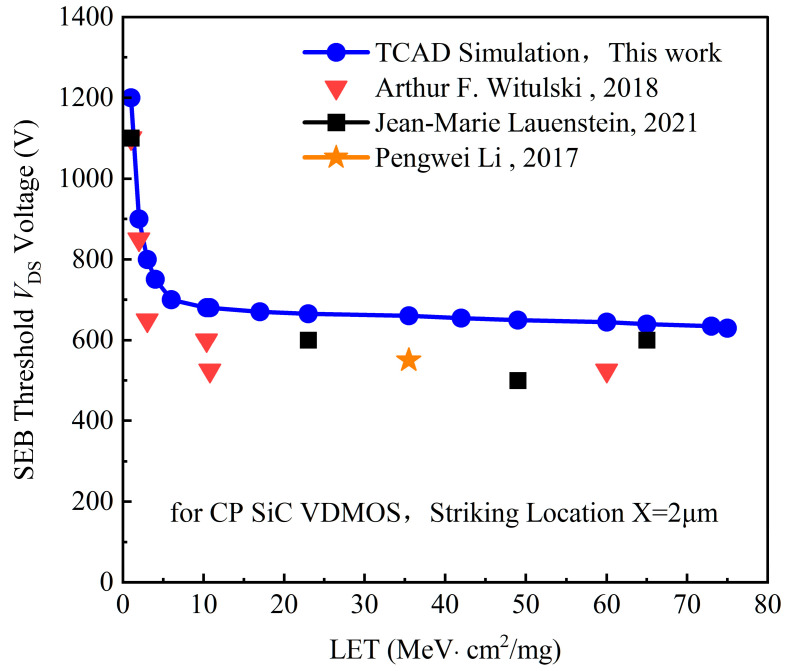
Comparison of the SEB threshold *V*_DS_ voltage for 1200 V CP SiC VDMOS between the TCAD simulation and the heavy ion experiment results: striking location of X = 2 μm for the simulation, with measured results from Witulki [5], Lauenstein [9,10], and Pengwei Li [15].

**Figure 8 micromachines-14-01074-f008:**
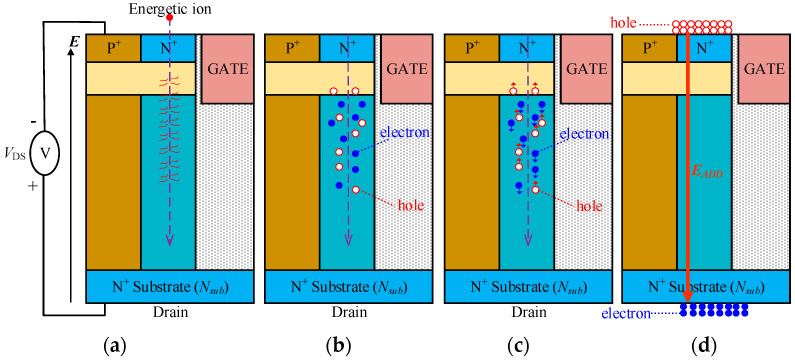
The concept of the SEE effect in VDMOS: (**a**) energy loss and transfer; (**b**) generation of e-h pairs; (**c**) e-h pairs drift; (**d**) electrons collected at the drain side and holes collected on the source side.

**Figure 9 micromachines-14-01074-f009:**
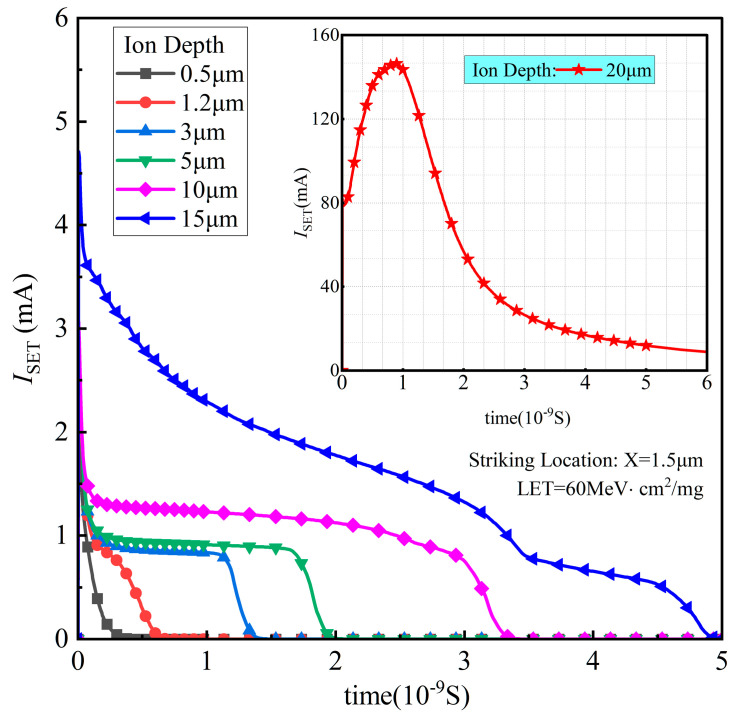
The DTSJ SiC VDMOS SET current simulation results under the conditions of a fixed drain bias voltage *V*_DS_ of 300 V, an ion striking location X = 1.5 μm, a LET of 60 MeV·cm^2^/mg, and a penetration depth ranging from 0.5 μm to 20 μm.

**Figure 10 micromachines-14-01074-f010:**
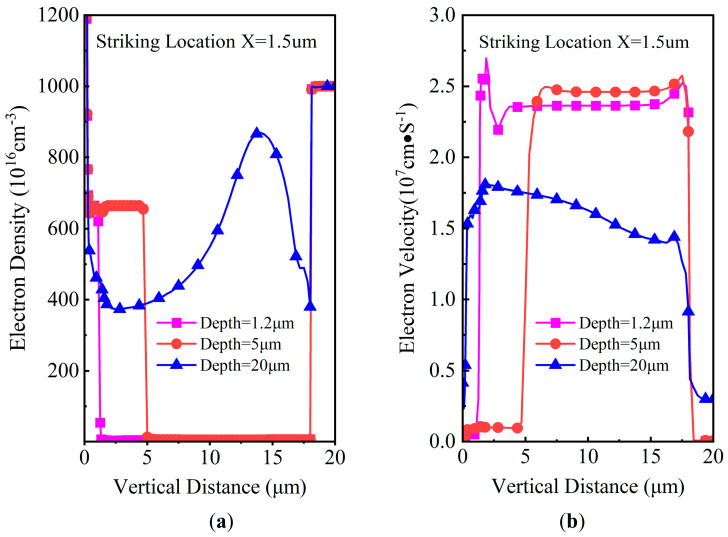
DTSJ SiC VDMOS cross-section of the electron properties in location X = 1.5 μm with different ion penetration depths of 1.2 μm, 5 μm, and 20 μm: (**a**) electron density; (**b**) electron velocity.

**Figure 11 micromachines-14-01074-f011:**
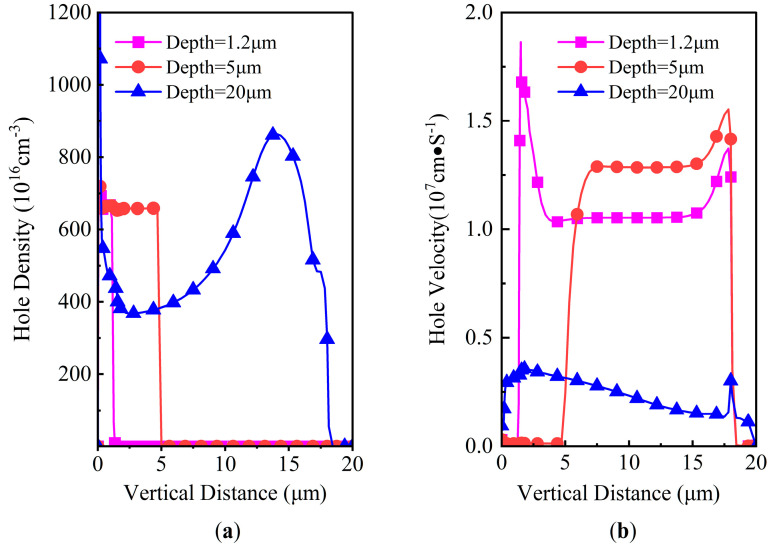
DTSJ SiC VDMOS cross-section of hole properties in location X = 1.5 μm with ion penetration depths of 1.2 μm, 5 μm, and 20 μm: (**a**) hole density; (**b**) hole velocity.

**Figure 12 micromachines-14-01074-f012:**
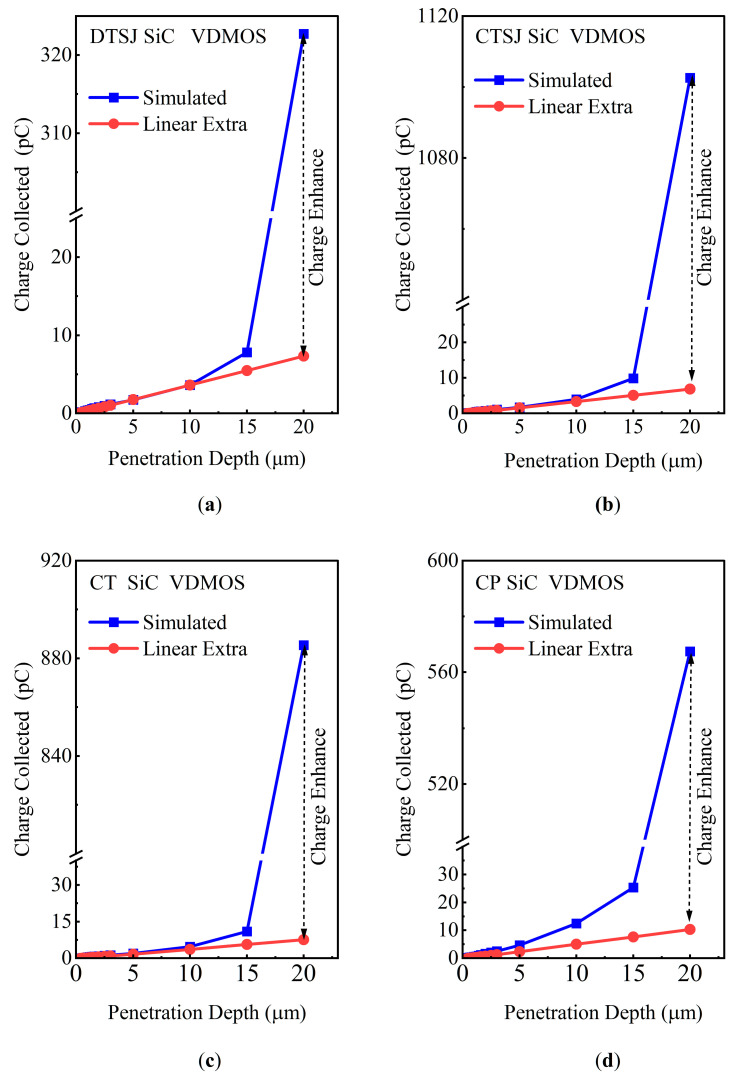
The quantity of charge collected at the drain, versus the heavy ion penetration depth, with an LET value of 15 MeV·cm^2^/mg of: (**a**) DTSJ SiC VDMOS; (**b**) CTSJ SiC VDMOS; (**c**) CT SiC VDMOS; (**d**) CP SiC VDMOS.

**Figure 13 micromachines-14-01074-f013:**
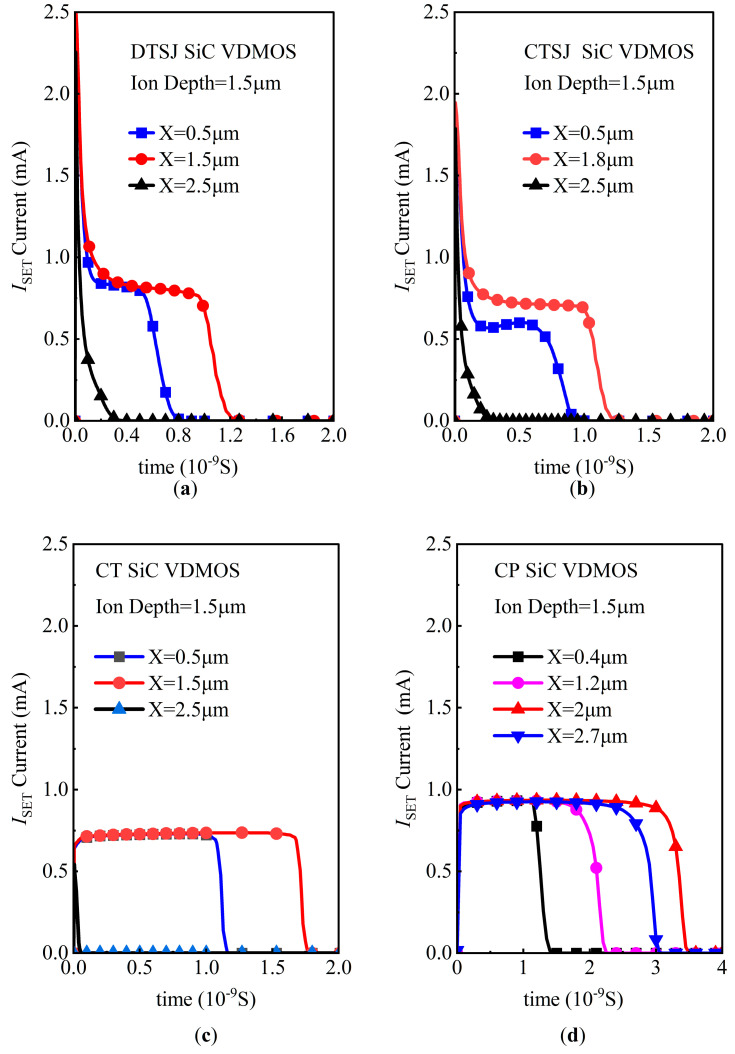
The SET current versus the different striking locations and LETs, with a fixed penetration depth of 1.5 μm and a drain bias voltage *V*_DS_ of 300 V for (**a**) DTSJ−, (**b**) CTSJ−, (**c**) CT−, and (**d**) CP SiC VDMOS.

**Figure 14 micromachines-14-01074-f014:**
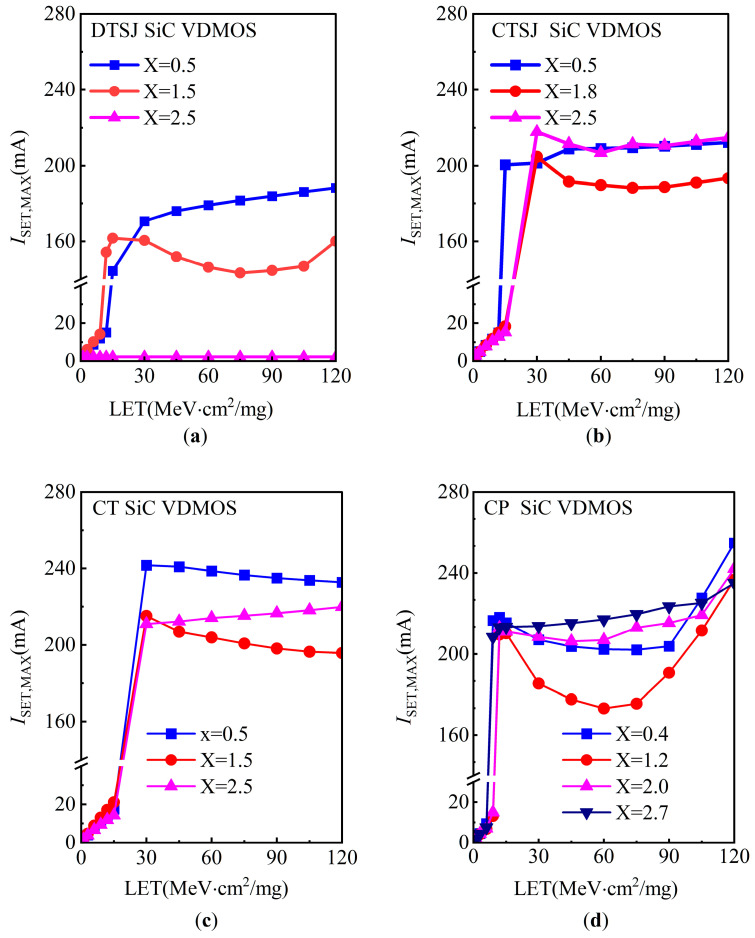
The SET current at different striking locations and LET, with a fixed penetration depth of 20 μm and a fixed drain bias voltage *V*_DS_ of 300 V for (**a**) DTSJ−, (**b**) CTSJ−, (**c**) CT−, and (**d**) CP SiC VDMOS.

**Figure 15 micromachines-14-01074-f015:**
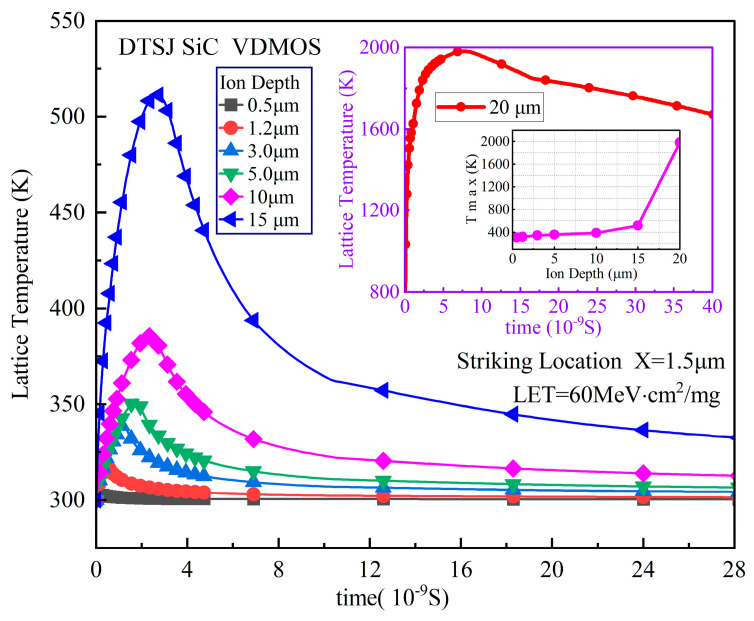
The DTSJ SiC VDMOS SET lattice temperature simulation results under the condition of an ion striking location of X = 1.5 μm, LET of 60 MeV·cm^2^/mg, a fixed drain bias voltage *V*_DS_ of 300 V, and ion penetration ranging from 0.5 μm to 20 μm.

**Figure 16 micromachines-14-01074-f016:**
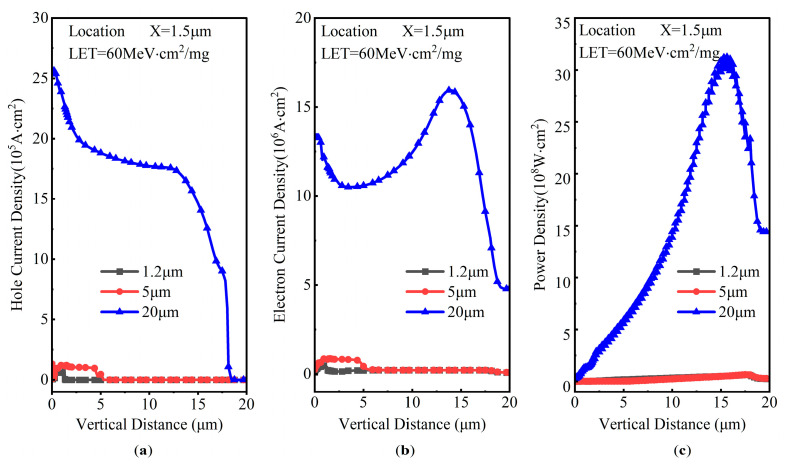
The DTSJ SiC VDMOS SET power distribution along the vertical distance, with a striking location X = 1.5 μm, LET of 60 MeV·cm^2^/mg, and fixed drain bias voltage *V*_DS_ of 300 V for (**a**) hole current density, (**b**) electron current density, and (**c**) power density.

**Figure 17 micromachines-14-01074-f017:**
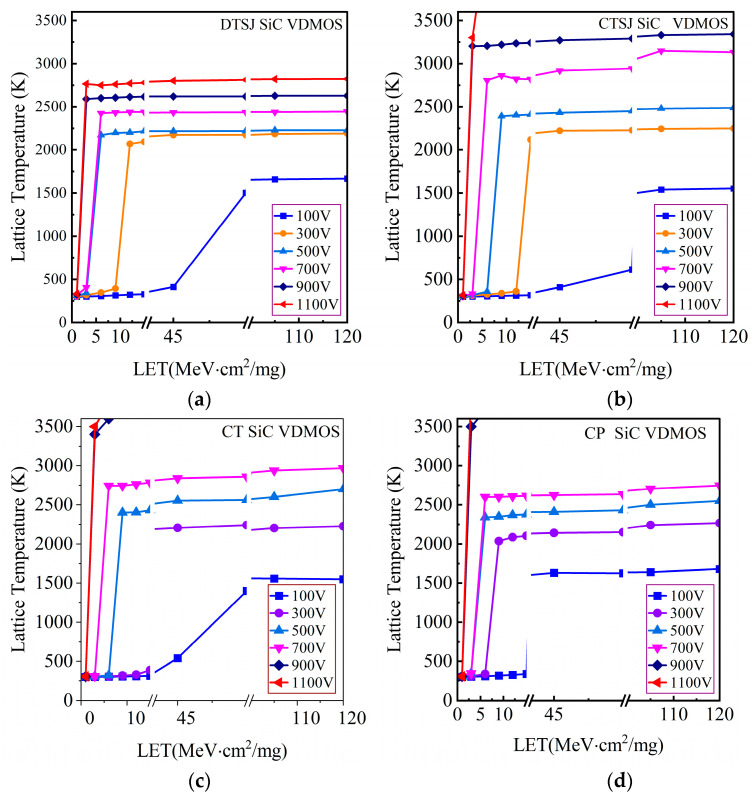
The maximum temperature matrix results, with *V*_DS_ ranging from 100 V to 1100 V with a step of 200 V and LET values ranging from 1 to 120 MeV·cm^2^/mg for (**a**) DTSJ−, (**b**) CTSJ−, (**c**) CT−, and (**d**) CP SiC VDMOS.

**Figure 18 micromachines-14-01074-f018:**
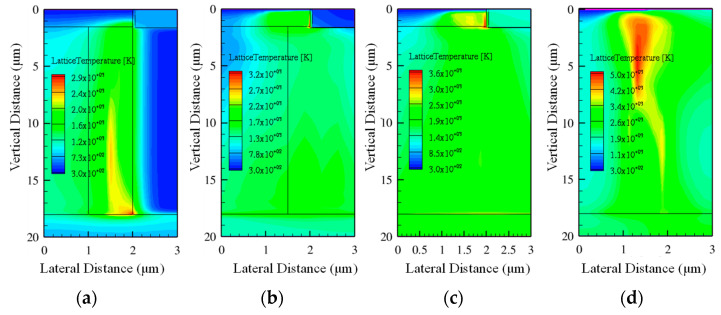
Temperature spatial distribution, where *V*_DS_ = 1100 V, LET = 120 MeV·cm^2^/mg, and location X = 1.5 μm in (**a**) DTSJ−, (**b**) CTSJ−, (**c**) CT−, and (**d**) CP SiC VDMOS.

**Figure 19 micromachines-14-01074-f019:**
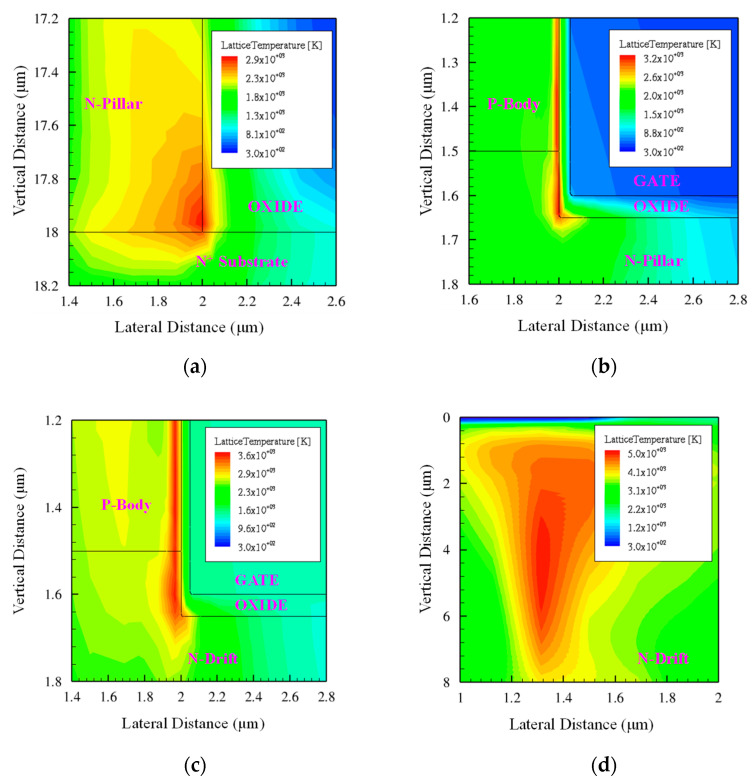
Temperature distribution in zoomed–in images from Figure 18 for (**a**) DTSJ−, (**b**) CTSJ−, (**c**) CT−, and (**d**) CP SiC VDMOS.

**Figure 20 micromachines-14-01074-f020:**
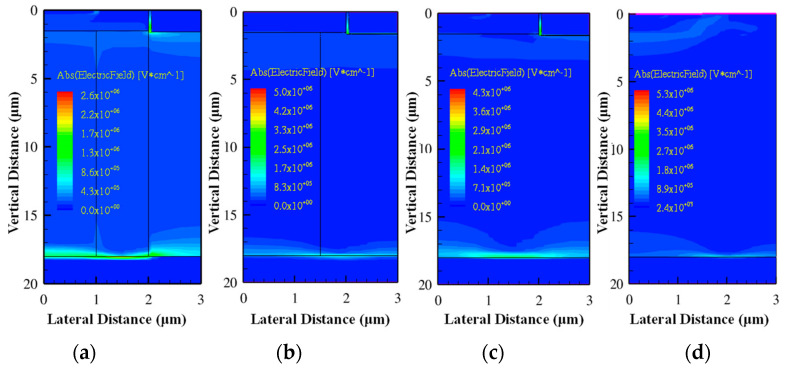
The electric field spatial distribution with *V*_DS_ = 300 V, LET = 60 MeV·cm^2^/mg, a time of 5 pS, and a heavy ion striking depth of 20 μm in (**a**) DTSJ−, (**b**) CTSJ−, (**c**) CT−, and (**d**) CP SiC VDMOS.

**Figure 21 micromachines-14-01074-f021:**
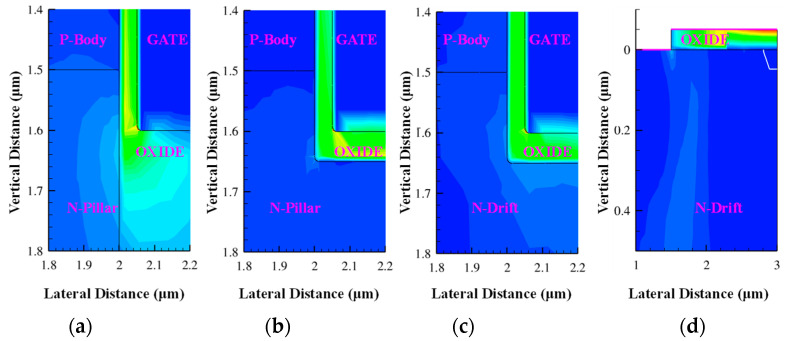
The electric field peak location of the zoomed–in Figure 20 for (**a**) DTSJ−, (**b**) CTSJ−, (**c**) CT−, and (**d**) CP SiC VDMOS.

**Figure 22 micromachines-14-01074-f022:**
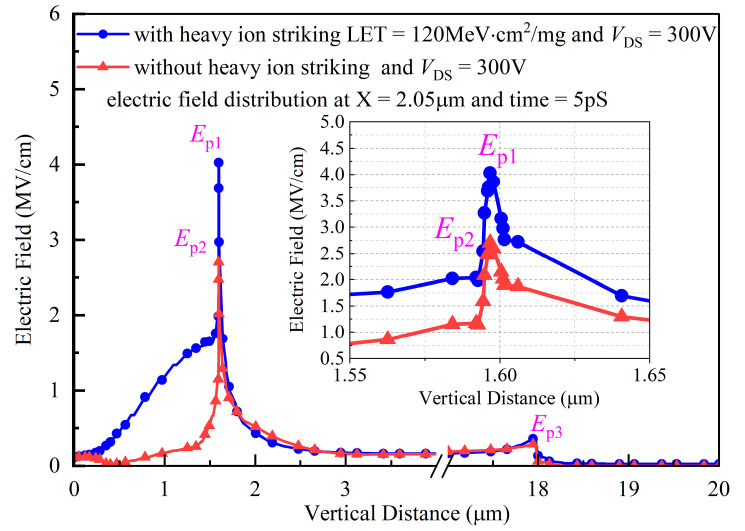
The comparison of electric field distribution along a vertical distance at X = 2.05 μm with heavy ion striking and without heavy ion striking for DTSJ SiC VDMOS.

**Figure 23 micromachines-14-01074-f023:**
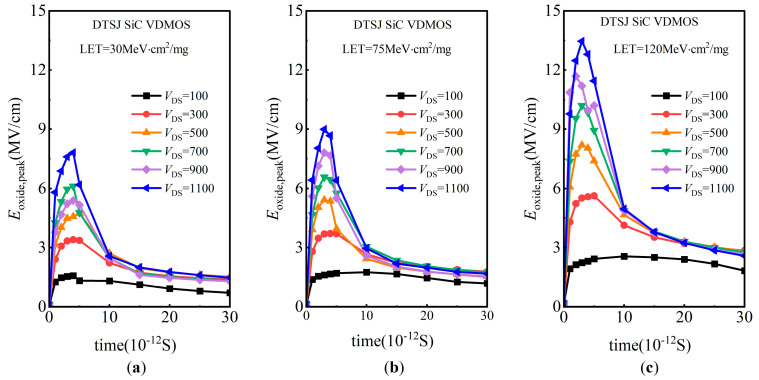
The evolution of the electric field peaks in the oxide region for DTSJ SiC VDMOS with different *V*_DS_: (**a**) LET = 30 MeV·cm^2^/mg (**b**) LET = 75 MeV·cm^2^/mg (**c**) LET = 120 MeV·cm^2^/mg.

**Figure 24 micromachines-14-01074-f024:**
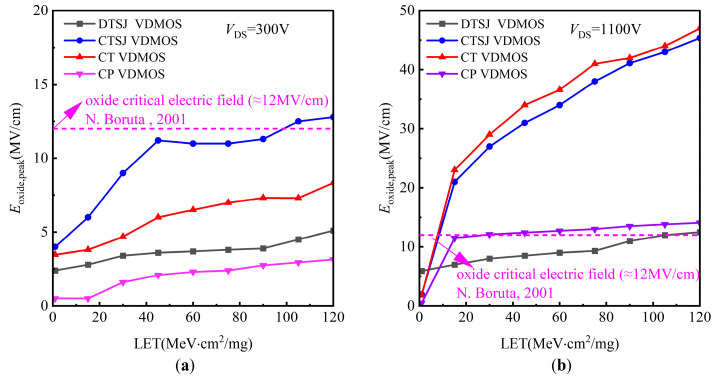
The electric field peaks in the oxide region for DTSJ−, CTSJ−, CT−, and CP SiC VDMOS, with LET values ranging from 1 MeV·cm^2^/mg to 120 MeV·cm^2^/mg: (**a**) *V*_DS_ = 300 V; (**b**) *V*_DS_ = 1100 V. N. Boruta, 2001 [25].

**Table 1 micromachines-14-01074-t001:** Structure parameters of the optimized SiC VDMOS devices.

Parameters	DTSJ−	CTSJ−	CT−	CP−
Width of half-cell (*W*, μm)	3	3	3	3
Thickness of oxide (*T*_ox_, μm)	0.05	0.05	0.05	0.05
Thickness of N-drift region (*T*_epi_, μm)	18	18	18	18
Thickness of P-body region (*T*_body_, μm)	1.5	1.5	1.5	1.5
Width of P-pillar region (*W*_sj_, μm)	1	1.5	—	—
Width of N-pillar region (*W*_sj_, μm)	1	1.5	—	—
Thickness of N+ substrate region (*T*_sub_, μm)	2	2	2	2
Doping concentration of P-body contact region (cm^−3^)	1 × 10^19^	1 × 10^19^	1 × 10^19^	1 × 10^19^
Doping concentration of N+ source region (cm^−3^)	1 × 10^19^	1 × 10^19^	1 × 10^19^	1 × 10^19^
Doping concentration of P_well_ region (P_well_, cm^−3^)	1 × 10^17^	1 × 10^17^	1 × 10^17^	1 × 10^17^
Doping concentration of N/P-pillar region (*N*_sj_, cm^−3^)	5.6 × 10^16^	2.8 × 10^16^	—	—
Doping concentration of N-drift region (*N*_d_, cm^−3^)	—	—	5.2 × 10^15^	7 × 10^15^
Doping concentration of N+ substrate (*N*_sub_, cm^−3^)	1 × 10^19^	1 × 10^19^	1 × 10^19^	1 × 10^19^

## Data Availability

Data available on request due to restrictions, e.g., privacy or ethical.

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
