# Peer review of "Simulation Studies on Single-Event Effects and the Mechanisms of SiC VDMOS from a Structural Perspective"

_micromachines, 2023, doi:10.3390/mi14051074_

Round 1

Reviewer 1 Report (Previous Reviewer 2)

The simulation setup is now better explained. The comparison on SET currents between the different MOS structures thanks transients simulations has been done and is well appreciated. 

Some typo remain (such as spacing between between values and units) especially in the newly added text.

Author Response

Thank you for your letter and for the reviewers’ comments concerning our manuscript entitled “Simulation Studies on Single Event Effects and Mechanisms of SiC VDMOS from the Structure Perspective” (micromachines-2369081).

Those comments are all valuable and helpful for revising and improving our paper, as well as the important guiding significance to our researches. We have studied comments carefully and have made correction which we hope meet with approval. Revised portion are highlighted in red in the paper. The corrections in the paper and the responds to the reviewer’s comments are as following:

Comments from the editors and reviewers:

Reviewer #1:

The simulation setup is now better explained. The comparison on SET currents between the different MOS structures thanks transients simulations has been done and is well appreciated.

Comments ï¼š

  1. Some typo remain (such as spacing between between values and units) especially in the newly added text.

For question1:

We are very grateful for your careful review and valuable suggestions. a space between value and unit is added in the full paper.at the same time, other typos have also been checked by grammar tool, read and revised by English professional.

Reviewer 2 Report (New Reviewer)

This paper presents a comprehensively numerical simulation study on single event effects for the proposed DTSJ SiC VDMOS and the other three conventional SiC VDMOS, uncovers the characteristics and mechanisms of single event transient and single event burn, especially, proposes innovatively a definition of the charge enhancement factor. But the paper still has some minor problems that need to be improved.

1. as mentioned in paper, single event effect characterization parameter LET_f is 0.07 times of LET, but the coefficient should be 0.007 from my knowledge?

2. as mentioned in paper, specific on resistance Ron,sp is defined as W×1μm×VDS/IDS , Ron,sp is closely related with VDS, it seems that VDS is critical, but the VDS for calculation of Ron,sp is not given?

3. The right bracket for the vertical coordinates in Figure 4(b) are marked with superscripts, this needs to be changed. In Figure 4(c) the brackets for both the horizontal and vertical coordinates are marked with full angle symbols and they need to be changed all the way to half angle symbols.

4. The units of both the horizontal and vertical coordinates in Figure 18 are written in um, which needs to be changed to μm. The full text needs to be checked to correct this error.

5. the electric field distributions of the devices must be provided, especially in the oxide and trench corner.

Author Response

Dear Editor and Reviewers:
Thank you for your letter and for the reviewers’ comments concerning our manuscript entitled “Simulation Studies on Single Event Effects and Mechanisms of SiC VDMOS from the Structure Perspective” (micromachines-2369081).
Those comments are all valuable and helpful for revising and improving our paper, as well as the important guiding significance to our researches. We have studied comments carefully and have made correction which we hope meet with approval. Revised portion are highlighted in red in the paper. The corrections in the paper and the responds to the reviewer’s comments are as following:

Comments from the editors and reviewers: 

Reviewer #2:

This paper presents a comprehensively numerical simulation study on single event effects for the proposed DTSJ SiC VDMOS and the other three conventional SiC VDMOS, uncovers the characteristics and mechanisms of single event transient and single event burn, especially, proposes innovatively a definition of the charge enhancement factor. But the paper still has some minor problems that need to be improved.
1. as mentioned in paper, single event effect characterization parameter LET_f is 0.07 times of LET, but the coefficient should be 0.007 from my knowledge?
2. as mentioned in paper, specific on resistance Ron,sp is defined as W×1μm×VDS/IDS , Ron,sp is closely related with VDS, it seems that VDS is critical, but the VDS for calculation of Ron,sp is not given?
3. The right bracket for the vertical coordinates in Figure 4(b) are marked with superscripts, this needs to be changed. In Figure 4(c) the brackets for both the horizontal and vertical coordinates are marked with full angle symbols and they need to be changed all the way to half angle symbols.
 4. The units of both the horizontal and vertical coordinates in Figure 18 are written in um, which needs to be changed to μm. The full text needs to be checked to correct this error.
 5. the electric field distributions of the devices must be provided, especially in the oxide and trench corner.

For question1: 
Thank you very much for your carefull review, we are sorry that the mistake in writing occurred due to our carelessness. The mistake has been corrected, see in the formula(3) in the revised version. actually , parameter LET_f is 0.007 times of LET, which is also used in our simulation model, as shown in Figure 5b.
For question2: 
The VDS for calculation of Ron,sp is 5V, which is clearly specified in the revised version, see in the Line.72: “and the VDS =5V is chosen based on the consideration of large signal condition in switching mode for VDMOS)”.
For question3: 
Thank you very much for your carefull review, the typos in Figure 4b and 4c have been corrected.
For question4: 
The units of both the horizontal and vertical coordinates in Figure 18 and Figure 19 have change from um to μm, the similar typos in the full text have also been checked and corrected.
For question5: 
Firstly, The prior work conducted the electric field analysis of DTSJ SiC VDMOS by the authors [20].
Secondly, the SEE electric field simulation and analysis have been conducted again, and added to the full text, see the section “3.6. SEGR and Electric Field Analysis”.

3.6. SEGR and Electric Field Analysis
The electric field is an essential parameter in the design and optimization of power devices, as does the SEE simulation of SiC VDMOS. The electric field distribution in DTSJ-, CTSJ-, CT- and CP SiC VDMOS are simulated and illustrated in Figure 20, with LET of 60 MeV•cm2/mg, drain bias voltage VDS of 300 V, heavy ion striking depth of 20 μm, and electric field captured at 5 pS after heavy ion striking.
The zoomed-in diagrams where the maximum electric fields occur are shown in Figure 21. the electric field peaks for three kinds of trench SiC VDMOS devices in oxide regions are located at the intersection of oxide and corner of the gate. the electric field peak for the CP SiC VDMOS is simply located at the side of oxide close to the gate.
The electric fields under heavy ion striking and without heavy ion striking with LET of 120 MeV•cm2/mg and drain bias voltage VDS of 300 V are compared and plotted in Figure 22. The heavy ion striking leads to the increase of electric field in the oxide region, the electric field peak Ep1 at 5 ps after heavy ion striking is about 4 MV/cm, the electric field peak Ep2 without heavy ion striking is about 2.54 MV/cm, the difference in magnitude between the two is about 1.6 times. The electric field in N-pillar and substrate regions for both with heavy ion striking and without heavy ion striking are almost the same. Moreover, another electric field spike Ep3 (≈0.6 MV/cm) exists at the intersection of N-pillar and substrate region in the case of heavy ion striking. 

(a)    (b)    (c)    (d)
Figure 20. The electric field spatial distribution with VDS=300 V, LET =60 MeV•cm2/mg, time of 5 pS and heavy ion striking depth of 20 μm in (a) DTSJ- (b) CTSJ- (c) CT- (d) CP SiC VDMOS.

(a)    (b)    (c)    (d)
Figure 21. The electric field peak location of the zoomed-in Figure 20 for (a) DTSJ- (b) CTSJ- (c) CT- (d) CP SiC VDMOS.

Figure 22. The comparison of electric field distribution along vertical distance at X=2.05 μm with heavy ion striking and without heavy ion striking for DTSJ SiC VDMOS .
Evolution of the electric field in the oxide region over time for DTSJ SiC VDMOS with LET of 30 MeV•cm2/mg, 75 MeV•cm2/mg, and 120 MeV•cm2/mg are shown in Figure 23a, Figure 23b, and Figure 23c, respectively. After heavy ion striking, the electric field increases immediately, and achieves the peak after several picoseconds, then decreases exponentially. The electric field peaks correlate linearly with LET and drain bias voltage VDS. The bigger the VDS and LET, the higher the electric field peaks. The electric field peak under LET=120 MeV•cm2/mg and VDS=1100 V is about 13MV/cm, which exceeds the critical electric field of oxide (≈12 MV/cm) [27], namely, SEGR occurs.
The electric field peaks in the oxide region for DTSJ-, CTSJ-, CT- and CP SiC VDMOS under the conditions of LET ranging from 1 MeV•cm2/mg to 120 MeV•cm2/mg and drain bias voltage of 1100 V are shown in Figure 24. As shown in Figure 24a, no event of SEGR happens for DTSJ-, CT- and CP SiC VDMOS when the drain bias voltage is less than 300 V, and the SEGR occurs for CTSJ SiC VDMOS when LET is larger than 100 MeV•cm2/mg. However, the probability of SEGR occurrence increases with LET and drain bias voltage VDS. As shown in Figure 24b, the SEGR LET thresholds are approximately 100 MeV•cm2/mg and 60 MeV•cm2/mg for DTSJ- and CP SiC VDMOS, respectively. By contrast, the electric field peaks for CTSJ- and CT SiC VDMOS significantly exceed the critical electric field of oxide, and the SEGR LET thresholds for CTSJ- and CT SiC VDMOS are about 15 MeV•cm2/mg.

Figure 23. The evolution of electric field peaks in the oxide region for DTSJ SiC VDMOS with different VDS (a) LET=30 MeV•cm2/mg (b) LET=75 MeV•cm2/mg (a) LET=120 MeV•cm2/mg.

Figure 24. The electric field peaks in the oxide region for DTSJ-, CTSJ-, CT- and CP SiC VDMOS with LET ranging from 1 MeV•cm2/mg to 120 MeV•cm2/mg (a) VDS=300 V (b) VDS=1100 V.

This manuscript is a resubmission of an earlier submission. The following is a list of the peer review reports and author responses from that submission.

Round 1

Reviewer 1 Report

This paper mainly studies the SEE effect and its mechanism of DTSJ silicon carbide VDMOS, and compares it with the other three structures. Through extensive simulation and analysis, it is verified that the proposed DTSJ silicon carbide VDMOS has the relatively best SET current and the lowest charge enhancement factor. The simulation experiments of the article are rich and the icons are clear, but there is a lack of experimental verification, and there is room for further improvement.

Comments :

1. The article has carried out a lot of simulation verification, but the description of the simulation environment parameters is not clear.

2. Through a large number of simulations, the proposed DTSJ silicon carbide VDMOS has the relatively best SET current and the lowest charge enhancement factor, but it lacks experimental verification. Can the feasibility of the proposed method be further verified by experiments ?

3. The analysis of the theoretical part is not sufficient, and the content of theoretical analysis in the second part of the article should be added to provide strong support for the research.

Author Response

Thank you for your letter and for the reviewers’ comments concerning our manuscript entitled “A Comparative Investigation on Single Event Effects and Mechanisms of SiC VDMOS from Structure Perspective” (micromachines-2278279).

Those comments are all valuable and helpful for revising and improving our paper, as well as the important guiding significance to our researches. We have studied comments carefully and have made correction which we hope meet with approval. Revised portion are highlighted in red in the paper. The corrections in the paper and the responds to the reviewer’s comments are as following:

Comments from the editors and reviewers:

Reviewer #1:

This paper mainly studies the SEE effect and its mechanism of DTSJ silicon carbide VDMOS, and compares it with the other three structures. Through extensive simulation and analysis, it is verified that the proposed DTSJ silicon carbide VDMOS has the relatively best SET current and the lowest charge enhancement factor. The simulation experiments of the article are rich and the icons are clear, but there is a lack of experimental verification, and there is room for further improvement.

Comments ï¼š

  1. The article has carried out a lot of simulation verification, but the description of the simulation environment parameters is not clear.
  2. Through a large number of simulations, the proposed DTSJ silicon carbide VDMOS has the relatively best SET current and the lowest charge enhancement factor, but it lacks experimental verification. Can the feasibility of the proposed method be further verified by experiments ?
  3. The analysis of the theoretical part is not sufficient, and the content of theoretical analysis in the second part of the article should be added to provide strong support for the research.

For question1:

The main parameter for SEE simulation is LET, LET characterizes the energy transferred into the device during heavy ion striking, LET is usually given by unit of pC/μm or MeV/mg/cm2, so there is no need to deal with other environmental parameters.

Physical models used in the simulation are given in Line.115-Line.118.

Heavy ion model used in the simulation is given in Line.119-Line.128, and illustrated in Figure 5.

For question2:

This work primarily focuses on simulation and theoretical analysis of SEE mechanisms of the proposed structure by simulation. Therefore, the paper tile is modified to “Simulation Studies on Single Event Effects and Mechanisms of SiC VDMOS from the Structure Perspective”.

The proposed structure has not been manufactured. The SEE experiments will be done absolutely in the future.

For question3:

This work mainly focuses on the SEE mechanisms study of the proposed structure. The prior work conducted the theoretical analysis of DTSJ by the authors [14].

The SET current mechanism is analyzed in Line.169-Line.185, Figure.8- Figure.10.

The charge enhancement mechanism is analyzed in Line.198-Line.211.

The SEB mechanism is analyzed in section “3.4. SEB and Thermal Analysis”, such as in Line.283-Line.300, Line.365-Line.371, etc.

Reviewer 2 Report

Can you specify the model names used for your TCAD simulations?

What is the lifetime of each of the structures presented?

The seconds unit is not correctly written. Use the lower case character. 

Fig.4b and c: please specify the cutline positions of the VDMOS structure as the structure seems to be two dimensionnal while the profile is given for a specific horizontal position.

For all the presented structures, have you estimated the impact of the doping concentrations for the most sensitive regions? Have you taken into account the traps that may be found at the oxide/SiC interface?

Does the current density in the drift layer play a major role in the lattice temperature increase? That would then easier to explain that increase with the current density.

Some typos should be corrected after review of the manuscript.

Author Response

Thank you for your letter and for the reviewers’ comments concerning our manuscript entitled “A Comparative Investigation on Single Event Effects and Mechanisms of SiC VDMOS from Structure Perspective” (micromachines-2278279).

Those comments are all valuable and helpful for revising and improving our paper, as well as the important guiding significance to our researches. We have studied comments carefully and have made correction which we hope meet with approval. Revised portion are highlighted in red in the paper. The corrections in the paper and the responds to the reviewer’s comments are as following:

Comments from the editors and reviewers:

Reviewer #2:

1.Can you specify the model names used for your TCAD simulations?

2.What is the lifetime of each of the structures presented?

3.The seconds unit is not correctly written. Use the lower case character. 

4.Fig.4b and c: please specify the cutline positions of the VDMOS structure as the structure seems to be two dimensional while the profile is given for a specific horizontal position.

5.For all the presented structures, have you estimated the impact of the doping concentrations for the most sensitive regions? Have you taken into account the traps that may be found at the oxide/SiC interface?

6.Does the current density in the drift layer play a major role in the lattice temperature increase? That would then easier to explain that increase with the current density.

7.Some typos should be corrected after review of the manuscript.

For question1:

The model used in the simulation has been updated. Physical models used in the simulation are given in Line.115-Line.118.

The heavy ion model used in the simulation is given in Line.119-Line.128 and illustrated in Figure 5.

For question2:

This paper mainly focuses on the SEE simulation study, so there is no need to involve with the lifetime of each structure.

For question3:

Thank you very much for your suggestions. as suggested, the Seconds unit uses the lower case character. See Figure 4a, Figure 8, Figure 12, and Figure 14.

For question4:

“Cross section x=1.2μm ” are specified in Fig.4b and c.

For question5:

According to simulation results, sensitive regions are correlated with device structure, doping and heavy ion striking positions.

Traps at the oxide-SiC interface fundamentally impact the threshold voltage and total dose irradiation effects, traps do not influence the single event effect.

For question6:

Surely. P=V×J, where P denotes power density, V denotes the voltage or potential, J denotes the current density. Obviously, the larger the current density, the higher power density, furtherly, the higher lattice temperature.

For question7:

Typos have been corrected in the full text, and English Professionals have read and revised the entire paper.

Reviewer 3 Report

1. The intruduction gives almost no information about experimental results on SEE problem and about the relevance of research. Among the cited papers only 3 form last 3-4 years. Others are older. Surely introduction shild be expanded and more relevant papers should be cited. The reader should be able to understansв where the key paramameters of the research are come from. 

2. Completely no explanation of the modelling is given. The structures are presented, but no modelling details are explained.

3. No conclusion or comparison with experimental results.

Taking into account the above, I believe that the work cannot be accepted without significant revision.

There are some comment on Eglish quality and quality of writting in general:

1. English should be improved by native speaker. Here I sum up style and grammar problems of 1st paragraph:

- line 40 "Shottkey", should be "Shottky"

- line 42 "etal", should be "et. al." 

- line 39, 40 words Heavy and Insulated should not start with capital letter

- line 34 "merits of ... impedance, ... switching, easily driven" -  the last term shuld be in the form of noun too.

 - line 44 "... has found occurences .... leakedge current in SiC VDMOS by heavy ion implantation" -  it is difficult to understand. Probably the authors meant that ion implantation leads to increase in leakedge current.

2. VDMOS, LET, FOM and other abbreviations should be explained not only in keywords but also in the text

Author Response

Thank you for your letter and for the reviewers’ comments concerning our manuscript entitled “A Comparative Investigation on Single Event Effects and Mechanisms of SiC VDMOS from Structure Perspective” (micromachines-2278279).

Those comments are all valuable and helpful for revising and improving our paper, as well as the important guiding significance to our researches. We have studied comments carefully and have made correction which we hope meet with approval. Revised portion are highlighted in red in the paper. The corrections in the paper and the responds to the reviewer’s comments are as following:

Comments from the editors and reviewers:

Reviewer #3:

  1. The introduction gives almost no information about experimental results on SEE problem and about the relevance of research. Among the cited papers only 3 form last 3-4 years. Others are older. Surely introduction should be expanded and more relevant papers should be cited. The reader should be able to understand where the key parameters of the research are come from. 
  2. Completely no explanation of the modelling is given. The structures are presented, but no modelling details are explained.
  3. No conclusion or comparison with experimental results.

There are some comment on English quality and quality of writing in general:

  1. English should be improved by native speaker. Here I sum up style and grammar problems of 1st paragraph:

- line 40 "Shottkey", should be "Shottky"

- line 42 "etal", should be "et. al." 

- line 39, 40 words Heavy and Insulated should not start with capital letter

- line 34 "merits of ... impedance, ... switching, easily driven" -  the last term shuld be in the form of noun too.

 - line 44 "... has found occurences .... leakedge current in SiC VDMOS by heavy ion implantation" -  it is difficult to understand. Probably the authors meant that ion implantation leads to increase in leakedge current.

  1. VDMOS, LET, FOM and other abbreviations should be explained not only in keywords but also in the text

For question1:

The research on SEE of SiC VDMOS started to gain a interest from academia and industry within the past ten years. In contrast to silicon power devices, investigations on SEE and mechanisms of SiC VDMOS are very few. Thus, the number of papers on SEE of SiC VDMOS is limited.

We reviewed the literature again on SEE of SiC VDMOS in recent 3-4 years and revised the literature citation in the introduction, see line.45-Line.46, References14, References15, References16.

DTSJ SiC VDMOS is the novel structure proposed by authors, and it has relatively better electric performance. Besides, the single event effect issue of SiC VDMOS is nowadays widely regarded as one of the most critical concerns in space applications. Therefore, the authors conducted the research work.

For question2:

This work mainly focuses on the SEE mechanisms study of the proposed structure; theoretical analysis and modeling of DTSJ have been conducted in the prior work by the authors[14].

The model of the charge enhancement factor is explained in Line.198-Line.211.

For question3:

DTSJ SiC VDMOS is the novel structure proposed by the authors. This work mainly focuses on simulation and theoretical analysis of SEE mechanisms of the proposed structure by simulation. Therefore, the paper title is modified to “Simulation Studies on Single Event Effects and Mechanisms of SiC VDMOS from the Structure Perspective”.

The proposed structure has not been manufactured until now. The SEE experiments will be done absolutely in the future.

For question4:

Thank you very much for your valuable comments. English Professionals has read and revised the paper.

Typo and grammar problems have also been corrected.

For question5:

Thank you very much for your valuable comments. The abbreviations of VDMOS, FOM, LET and other abbreviations have been explained where they appear in the context for the first time. See Line.33-34, Line.70, Line.98.

Round 2

Reviewer 2 Report

The manuscript has been amended accordingly to comments.

Author Response

Thank you for the comments concerning our manuscript entitled “A Comparative Investigation on Single Event Effects and Mechanisms of SiC VDMOS from Structure Perspective” (micromachines-2278279). We have imroved our manuscript further.

Reviewer 3 Report

The authors of the work have made some changes, and certainly the article has become much better organized, but the main drawback remains. A hypothetical device that has not been manufactured is considered. Of course, pure modeling cannot predict all the technological difficulties and features in the implementation of the device. In my opinion, there are no experimental results so far, it is difficult to talk about the relevance of this topic.

It should be noted that the subject of the work is far from the scope of the journal. After careful consideration, I would recommend that authors consider publishing in another journal.

Author Response

First of all, I am so grateful for your valuable comment.

I would like to make some explanations.

Firstly, as emerging wide bandgap power devices, SiC MOSFET power device is gaining more and more attention in many fields, such as automotive and aerospace. Like Si MOSFET power device, people strive to improve the Figure of Merit (BV2/Ron,sp) of SiC MOSFET through the superjunction and trench approaches. Therefore, we have proposed a deep trench super-junction structure SiC VDMOS.

Secondly, the single-event effect of SiC MOSFET is a worldwide challenge. In contrast to silicon power devices, experimental and theoretical researches on SEE and the mechanisms of SiC VDMOS are relatively few in general. Although the proposed deep trench super-junction structure SiC VDMOS has not been manufactured, it is necessary to study theoretically in advance.

Thirdly, based on reviewer’s question, we simulate the SEB threshold VDS voltages for 1200V CP SiC VDMOS again. Then the simulated results are compared with heavy ion experiment results by Witulki [9], Lauenstein [10-11] and Pengwei Li [16], to validate the TCAD model. The details are as follows.
